# Gradual positive and negative affect induction: The effect of verbalizing affective content

**Charlotte Out** *, **Martijn Goudbeek** , **Emiel Krahmer**

Department of Communication and Cognition, Tilburg School of Humanities and Digital Sciences, Tilburg University, Tilburg, The Netherlands

* c.out@uvt.nl

## Abstract

In this paper, we study the effect of verbalizing affective pictures on affective state and language production. Individuals describe (Study I: Spoken Descriptions of Pictures) or passively view (Study II: Passively Viewing Pictures) 40 pictures for the International Affective Picture System (IAPS) that gradually increase from neutral to either positive or negative content. We expected that both methods would result in successful affect induction, and that the effect would be stronger for verbally describing pictures than for passively viewing them. Results indicate that speakers indeed felt more negative after describing negative pictures, but that describing positive (compared to neutral) pictures did not result in a more positive state. Contrary to our hypothesis, no differences were found between describing and passively viewing the pictures. Furthermore, we analysed the verbal picture descriptions produced by participants on various dimensions. Results indicate that positive and negative pictures were indeed described with increasingly more affective language in the expected directions. In addition to informing our understanding of the relationship between (spoken) language production and affect, these results also potentially pave the way for a new method of affect induction that uses free expression.

## 1. Introduction

Speaking about emotionally meaningful events is not a neutral act: it affects us. When we experience something positive, like getting a raise, we feel happy, and, conversely, explaining that we did not get a promotion makes us feel sad. This relationship between speaking and feeling is partly there because the event we talk about is inherently affective, but it might also be because verbalizing an emotionally meaningful fact ("I got a raise. I finally got a raise!") amplifies or even induces the emotions we experience.

In this paper, we investigate the relationship between language production and affect. Inspired by the Velten method [1], we present a procedure where participants are exposed to pictures that increase or decrease in valence as the experiment progresses. In contrast to the Velten method that relies on reading aloud affective fixed self-referential statements, we use

**Data Availability Statement:** The datasets and syntax can be found in the folder "Gradual emotion induction with a visual Velten method" –> "Datasets and syntax" at the Open Science Foundation: https://osf.io/7zgct/

**Funding:** This study was funded by the Netherlands Organization for Scientific Research [360-89-050]. The award was granted to EK (principal investigator) and Mariët Theune and MG (co-applicants). The funders had no role in study design, data collection and analysis, decision to publish, or preparation of the manuscript. More information on the project can be found at https://www.nwo.nl/onderzoek-en-resultaten/onderzoeksprojecten/i/45/13545.html.

**Competing interests:** The authors have declared that no competing interests exist.

spontaneous spoken descriptions of affectively charged pictures. To elucidate the relationship between affective state and language production, we explore the content of the verbal descriptions, using the affective categories of the Linguistic Inquiry and Word Count (LIWC; [2]). To gauge the role of verbal description in eliciting affective reactions, we explore the difference between describing these pictures and silently viewing them.

## 1.1. The Velten method

The theoretical aim of Velten's [1] study was to find evidence for the efficacy of a type of cognitive therapy, focusing on making the patient aware of how their own verbal interpretations of events influence their affective responses. Predicting that affective phrases would indeed elicit corresponding responses, Velten created an affect induction method to elicit positive (elation) or negative (depressive) affect by having participants read out loud 60 sentences that gradually increased in affective content. The first sentence in both conditions was 'Today is neither better nor worse than any other day'. In the positive condition, it was followed by sentences such as 'Things look good, things look great!' with the final sentence being 'God, I feel great!'. In the negative condition, it was followed by sentences such as 'It often seems that no matter how hard I try, things still go wrong', with the final sentence being 'I want to go to sleep and never wake up'. Velten also included a neutral condition, containing sentences as 'The review is concerned with the first three volumes' and 'West Samoa gained its independence in 1965'. After reading the statements, various measures were obtained, including several cognitive and behavioral tasks. In one of the tasks, participants were asked to choose from a long list of adjectives which adjectives applied to them (Multiple Affect Adjective Check List, Today Form (MAACL) [3], and the experiment leader kept track of the number of words the participant uttered during the tasks.

As Velten predicted, participants who read the negative statements, compared to those who read the positive statements, ticked significantly more adjectives in the Depression Scale, one of the five emotion subscales of the MAACL, and in general uttered less words [4]. Velten concluded that these results, together with his other measurements, indicated that the induction method was effective: participants reading the negative statements felt more depressed, and participants reading positive statements felt more elated. Participants reading the neutral statements generally fell between the scores of elation and depression, implying that no effect on emotion took place [1]. After its initial development, the Velten method has been widely used in the following decades (e.g., [5]), and the effects have frequently been replicated (e.g., [5–7])

## 1.2. The relationship between verbal expression and affective state

In general, the Velten method appears to be highly relevant for researchers studying the relationship between affect and language, which has been under scholarly debate for only about a decade [8], resulting in various hypotheses. One hypothesis is based on the psychological constructionist approach. According to this approach, language has a pivotal, although not sufficient, role in perceiving and experiencing emotions. The approach suggests that the lexicon of (affective) words at our disposal is essential to make meaning of, and therefore shape, our affective experiences, for example, by turning general, vague feelings of displeasure ('this doesn't feel right') into concrete emotions ('I feel lonely', [8]). According to Barrett [9], this categorization is learned from infancy, and depends on the social (and cultural) environment. To a certain extent, emotions are *created* by naming, and therefore categorizing (and experiencing) them.

However, the literature shows conflicting results with respect to the impact of verbalization on affective state. On the one hand, there is support that expressing affective content can

attenuate the affective experiences, e.g., via 'affect labelling'. When individuals use affect label-ling and put their emotions into words this can result in a decrease in the intensity of the affec-tive, often negative, experience (see, e.g., [10–12]). Therefore, some consider affect labelling to be an unintentional or incidental form of affect regulation (see e.g., [13]). In one study, Fan et al. [10] studied affect labelling in naturally occurring, spontaneous emotional expression on Twitter, looking at tweets starting with 'I feel. . .', followed by an adjective or adverb, written by approximately 74.487 different Twitter users. A dictionary based affect detection algorithm, VADER (Valence Aware Dictionary and Sentiment Reasoner; [14]) was used to detect possible changes in affective content of the tweets six hours after (and before) the affect labelling took place. Their results showed that for most individuals, immediately after using affect labelling in a tweet, the level of affective content of their tweets decreased, before returning to baseline [10]. Negative emotions returned to baseline fairly rapidly, with a decay half-life of five min-utes, while for positive emotions a less rapid reduction was observed with a decay half-life of eleven minutes. The authors conclude that their findings are in line with literature on the attenuating effects of affect labelling.

In contrast to the findings summarized above, putting emotions into words can also result in an *enhanced* affective experience. For example, Ortner [15] presented participants with neu-tral and negative pictures. First, they merely viewed 10 neutral and 10 negative pictures. Then, for the next 10 neutral and 10 negative pictures, participants were asked to either passively view, reappraise (reinterpret the pictures in a way that it no longer seemed negative) or emo-tionally label pictures (observe which emotions they experience and utter their labels, e.g., 'there is. . . anger'). The results showed that participants using affect labelling reported stronger affective states than those reappraising or only viewing the pictures. Ortner [15] suggests that the individuals who verbally described their emotional reactions to the affective pictures in their own words created a heightened awareness of them and therefore, experienced more intense affect.

Finally, in expressive writing, the verbalizing of emotions often results in an initial increase, followed by a decrease in emotional intensity. Expressive writing is a well-known and success-ful technique to deliberately reduce (unwanted) negative affect and distress in the long run (e.g., [16, 17]). For this technique, individuals are asked to write 15–20 minutes about a trau-matic or personally emotional event for several consecutive days [18]. In contrast to Fan et al. [10], the decrease in negative emotions was not immediate: during, and immediately after writing, individuals usually reported feeling worse (e.g., [16,19]), although at least one study reports that individuals scored higher on positive disposition shortly after the last expressive writing episode [17]. In all these studies, participants experienced a decrease in negative affect in the long run, even while feeling worse immediately after the linguistic expression of the emotional event.

## 1.3. Describing affective pictures

The findings discussed above indicate the existence of a crucial, although unclear, relationship between language production and affective state and vice versa, especially when individuals are allowed to use their own words. Asking participants to describing affectively laden pictures might be an excellent way to bring about an affective state, while simultaneously allowing the free production of linguistic content that, in turn, might affect the extent to which the pictures induce an affective state compared to a situation where participants are merely passive observ-ers of the pictures.

For our study, we selected pictures from a well-known and validated set of affectively laden pictures, the International Affective Picture System (IAPS; [20, 21]). The IAPS is a large set of

pictures of varied content, including arousing and (un)pleasant (e.g., snakes and spiders, romantic couples, and extreme sports) as well as more neutral pictures (e.g., flowers, objects, and portraits). The pictures have been rated on valence or pleasure (negative/positive), arousal (low/high) and dominance (dominated/in control; [20]), as well as for discrete emotion categories [22].

Most studies using the IAPS pictures to induce affective state select a number of IAPS pictures within certain ranges of valence to create subsets of positive, negative and sometimes neutral pictures (e.g., [23]). Per the Velten method, we aimed to gradually induce positive and negative affect by exposing individuals to sets of IAPS pictures that start with neutral content, and gradually become more positive or negative, based on their valence ratings [20].

## 1.4. The current studies

In order to study the effect of language production on affect, we devised a method where individuals expressed themselves in their own (possibly affective) language as a way to induce affect. Our method is inspired by the incremental nature of the Velten method, but asks participants to describe affectively evocative pictures instead of read out loud sentences. This modification is prompted by the desire to investigate the relationship between language and affect, while simultaneously using a more natural paradigm that is less prone to demand characteristics.

To assess whether verbally describing the pictures would indeed result in an enhanced affective experience we contrast the effect of this with the effect of passively viewing pictures, which is the more conventional way of using the IAPS pictures (e.g., [23]). To our knowledge, this is the first study that investigates the effect of describing affectively charged pictures on the affective state of the speaker. Given that previous work found that describing personal affective experiences either decreases (e.g., [12]), increases (e.g., [15]), or increases and then decreases [16] the intensity of the experienced affect, this comparison could go either way. However, given that the verbalizing self-referential statements were effective in the Velten method, we hypothesize that, compared to passively viewing them, verbally describing affectively laden pictures will enhance the affective experience.

In sum, to investigate our research questions, we conduct two studies investigating whether there is an additive effect of verbalizing the content of pictures on affective state (compared to merely viewing them). For this, we used pictures taken from the IAPS, gradually increasing in affective content (positive, negative) or remaining neutral. In Study I (called "Spoken Descriptions of Pictures"), participants view and describe the pictures out loud. In Study II (called "Passively Viewing Pictures"), participants passively view the pictures, and do not describe them.

Finally, to elucidate the relationship between affective state and the language that is used in the descriptions, we will explore the content of the verbal descriptions of the pictures of Study I, comparing the frequency of affective word use in the three (affective) content categories, and word count, using LIWC [2]. We preregistered the methods, hypotheses, and analyses of this study at the Open Science Foundation: https://osf.io/kv8g3.

## 1.5. Hypotheses

With respect to our gradual affect induction procedures, we have the following hypotheses:

1. Irrespective of whether they describe the pictures out loud, participants in the condition with positive pictures will report higher levels of pleasant affect. In the condition with

negative pictures, they will report higher levels of unpleasant affect. No differential effect is expected for the neutral pictures.

2. We expect that *describing* affective pictures will enhance the effect on affective state compared to passively viewing the pictures. Specifically, we predict that participants viewing and describing positive pictures (Study I) will report higher levels of pleasant affect than participants passively viewing positive pictures (Study II), and participants viewing and describing negative pictures (Study I) will report higher levels of unpleasant affect than participants passively viewing negative pictures (Study II). To determine if this hypothesis is true, the results from Study I and Study II will be compared.

## 2. Study I: Spoken Descriptions of Pictures

Study I investigated the effect of viewing and describing (out loud) pictures gradually increasing in affective content on (self-reported) affective state.

### 2.1. Method

**2.1.1. Design.**   The study had a 2 (Time: pre-test, post-test) x 3 (Condition: positive, neutral, and negative) design, with time as within-subjects variable and condition as between-subjects variable.

**2.1.2. Participants.**   In total, 122 participants were recruited at a Dutch university and participated in the experiment for course credit. One participant was excluded because they did not consent to their data being used. Our final sample included $N$ = 121 participants (41 male; *M* age = 22.22, *SD* age = 2.90), each randomly assigned to one of the conditions (positive condition: $n$ = 41; neutral condition: $n$ = 40; negative condition: $n$ = 40).

All procedures performed were in accordance with the ethical standards of the institutional research committee, the Research Ethics and Data Management Committee of Tilburg School of Humanities and Digital Sciences, Tilburg University. All participants gave written informed consent in accordance with the Declaration of Helsinki (1964) and its later amendments or comparable ethical standards.

**2.1.3. Materials.**   *Stimuli*. We used the 2008 variant of the IAPS, containing 1194 pictures [24]. In order to create three conditions, we selected 40 IAPS pictures per condition based on the procedure described below.

First, two sets of 600 positive and negative pictures each were created. For the positive condition, we started from the 600 pictures with the highest valence rating (range 5.22–8.34). For the negative condition, we started from the 600 pictures with the lowest valence ratings (range 1.31–5.24). Indeed, the ranges of positive and negative pictures partly overlap. This mirrors the Velten method, which starts with the same sentence in both the positive and negative set of statements.

Next, both sets were divided into 40 bins of fifteen pictures, with each bin increasing in pleasant (positive condition) or unpleasant (negative condition) content. From each bin, one picture was randomly selected, resulting in two sequences of 40 pictures, which gradually increased in (un)pleasant content.

To create the picture set for the neutral condition, we selected 301 pictures with an average valence rating (range 4.62–5.92). Forty random bins of fifteen pictures were created, and from each bin one picture was randomly selected.

While selection of pictures from bins was random in principle, sometimes a picture was deemed inappropriate and replaced by another randomly selected picture from the same bin.

Exclusion criteria were: erotic or sexually suggestive (but not non-erotic nudity), too gruesome or disgust-inducing, repetitive content, or culturally sensitive content (e.g., traditions and rituals). Based on these criteria, we excluded six pictures and replaced them with more appropriate pictures from the same bin (S1 Table). Our final sample can be found in S2 Table.

Finally, to check if the three sets of pictures did not contain any outliers that would disturb the gradual increase (positive and negative condition), or would interfere with a consistent level (neutral condition) of affective content of the pictures, the sets of pictures were inspected for their valence and arousal in two line plots (S1 Fig and S2 Fig). As shown in S1 Fig, both the positive set, and the negative set, displayed a near perfect gradual increase in affective content in the expected directions. For the neutral set, a (very) slight decrease in pleasant content can be observed. As can be found in S2 Fig, compared to the valence ratings in S1 Fig, the arousal ratings were less distinct in their sequential direction, with a strong increase in arousal for negative pictures, and no substantial in- or decrease for both the positive pictures, as the neutral pictures.

For the three sets of pictures, the range, mean (with standard deviation) and median of the valence and arousal scores can be found in Table 1.

*Viewing IAPS pictures*. In Study I, participants were given instructions to describe each picture out loud, inspired by the MS COCO instructions [25], which is a well-established method of eliciting picture descriptions. In our study, participants were instructed to describe all the important aspects and details of the pictures, describe them in a way that another person could recognize this picture out of the set of 40 pictures, and use full sentences when describing the pictures.

After piloting with various timeframes (6, 8, and 10 seconds), 10 seconds viewing time per picture appeared to be sufficient to describe the pictures. Each participant started with two practice trials describing two neutral pictures. In order to encourage participants to actively engage in the task, we presented them with a bogus purpose of the study: memorizing the pictures. The study was introduced as a memory experiment, and participants were told that they would be asked to indicate pictures they had, and had not, seen before from a set of new and old (already seen) pictures.

*Video- and audio recording*. Audio was recorded for content analysis of the picture descriptions. In addition, we video recorded facial expressions for possible future analysis.

*Affect questionnaire*. Before and after viewing the series of pictures, participants indicated their current affective state on six 7-point Likert scales: sad/happy, unpleasant/pleasant, unsatisfied/satisfied, discontent/content, sullen/cheerful, low-spirited/in high spirits ([26], based on [27]; [28]; English translations of Dutch originals). They were instructed to choose a number per scale; the closer the numbers were to the words, the stronger they match the feeling described the word in question. Low numbers indicated the degree of negative affect (e.g., unpleasant), high numbers indicated the degree of positive affect (e.g., pleasant). In a previous study by Krahmer et al. [26], the internal consistency of this questionnaire was good, $\alpha = .88$. We assessed the reliability of the current scale with Cronbach's $\alpha$ as well. This analysis indicated that the items of our affect questionnaire had excellent internal consistency, for both Study I (pre-test, $\alpha = .90$; post-test, $\alpha = .94$) and Study II (pre-test, $\alpha = .94$; post-test, $\alpha = .95$). Based on these results, the six items were merged into one scale, 'Affect', resulting in one pre-test and one post-test score per participant, indicating (self-reported) affective state, ranging from 1 (negative affect) to 7 (positive affect). Based on these results, the six items were merged into one scale, 'Affect', resulting in one pre-test and one post-test score per participant, indicating (self-reported) affective state, ranging from 1 (unpleasant) to 7 (pleasant).

*Procedure*. After participants signed the informed consent form, the experiment leader explained the procedure and turned on the camera, including audio recording. If needed, the

**Table 1. Statistical characteristics of the final sample of pictures.**

| Pictures | Valence | | | Arousal | | |
|---|---|---|---|---|---|---|
| | Range | *M* (*SD*) | Median | Range | *M* (*SD*) | Median |
| Positive | 5.22–8.05 | 6.52 (0.77) | 6.52 | 2.63–7.31 | 4.33 (1.04) | 4.07 |
| Neutral | 4.95–5.22 | 5.08 (0.82) | 5.07 | 2.00–6.23 | 3.52 (0.93) | 3.22 |
| Negative | 1.51–5.22 | 3.53 (1.13) | 3.57 | 1.72–7.07 | 4.73 (1.55) | 4.97 |

camera was adjusted to an appropriate height to record the participant's face. Participants reported their gender and age. They then filled out the affect questionnaire for the first time (pre-test). Then, starting with two practice trials, participants were asked to view and describe 40 pictures out loud. After the task, they filled out the affect questionnaire again (post-test). Then, participants were asked to indicate which pictures they had seen before, and which ones they had not. Pictures were selected beforehand, by randomly picking three numbers between 1 and 40, using the corresponding bin to select one 'old', and one 'new' picture. Ninety percent (*n* = 220) of all participants (*N* = 245) correctly identified all six pictures as 'old' or 'new'.

After the experiment, participants in the negative condition viewed a light-hearted, short video displaying a jumping competition for bunnies [29]. This video was shown to rise their spirits, in case participants felt especially low after the experiment. Participants in the positive and neutral condition did not watch the video. At the end, the participants were debriefed and thanked for their participation.

## 2.2. Results

**2.2.1. Descriptive statistics.**    Fig 1 displays the individual scores on affect (y-axis) for the pre- and post-test (x-axis), sorted by condition (positive, neutral, and negative pictures). On the y-axis, lower scores indicate the degree of unpleasant affect; higher scores indicate the degree of pleasant affect. In Fig 1, the results for the three conditions show a clear pattern. Participants viewing negative pictures generally report feeling unpleasant after describing the pictures. Participants viewing positive pictures generally report feeling slightly more pleasant describing the pictures, and participants viewing the neutral pictures did not seem to report a change in affective state. In general, participants in all conditions seem to start the experiment in fairly good spirits (possibly partly explaining the limited effect in the positive condition), scoring roughly 5 to 5.5 on the 7-point Likert scale.

**2.2.2. Change in affective state.**    We performed a repeated measures analysis of variance with time (pre-test and post-test) as within-subjects factor, condition (positive, neutral, or negative pictures) as between-subjects factor and affective state as dependent variable. Mean scores, standard deviations, difference scores (posttest-pretest) and range can be found in Table 2. A main effect was found for time, $F(1, 118) = 28.03$, $p < .001$, $\eta_p^2 = .19$, and for condition, $F(2, 118) = 8.05$, $p = .001$, $\eta_p^2 = .12$. However, these two main effects were qualified by a (predicted) interaction effect for time and condition, $F(2, 118) = 37.23$, $p < .001$, $\eta_p^2 = .39$. Post-hoc tests revealed that participants in the negative condition reported lower levels of pleasantness after viewing and describing the negative pictures. Participants in the positive or neutral condition did not report a significant change in affective state after viewing and describing the pictures.

## 2.3. Conclusion

As predicted, participants viewing and describing negative pictures reported to experience lower levels of pleasantness, and participants viewing and describing neutral pictures did not

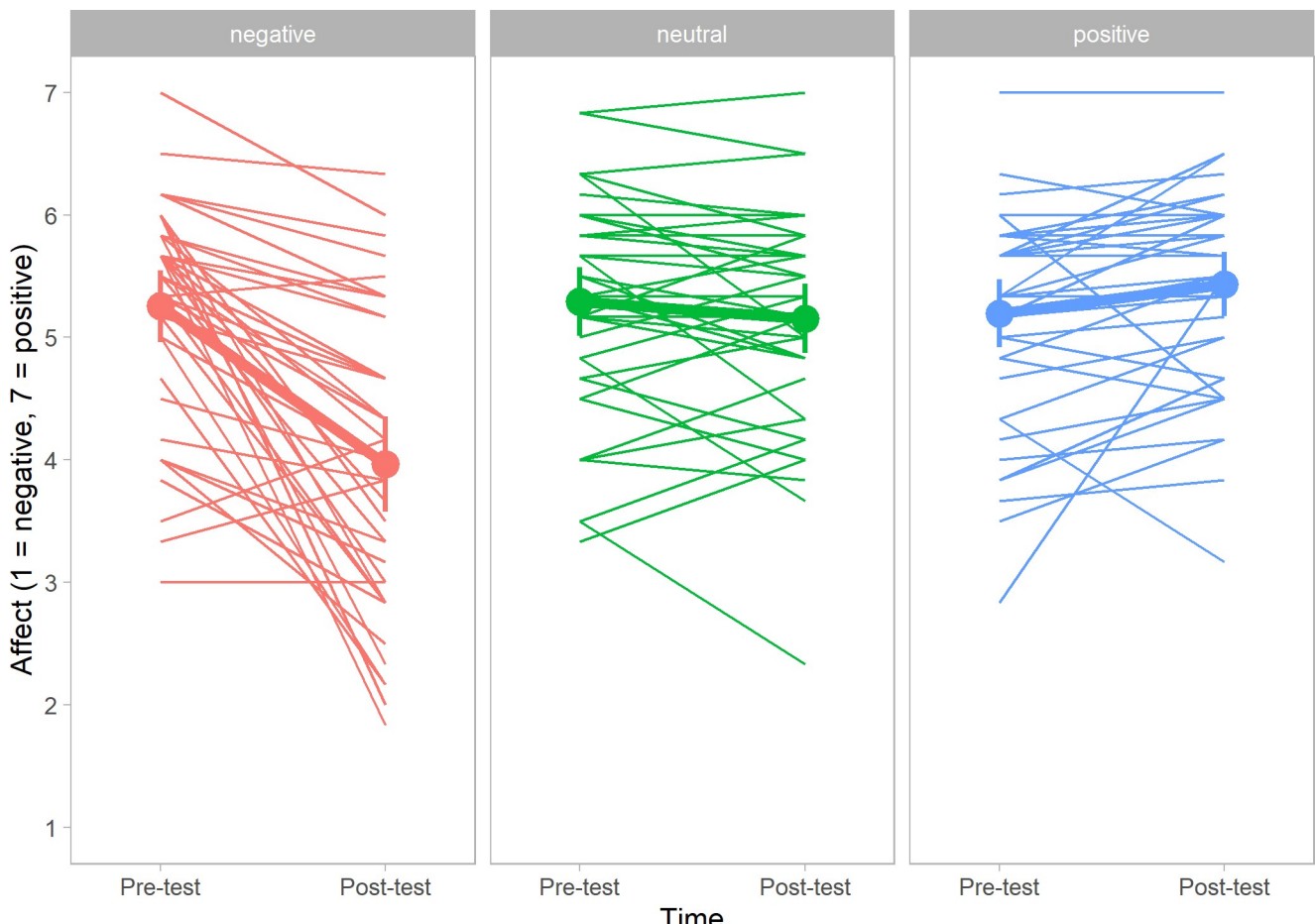

**Fig 1. Individual affect scores for participants viewing and describing pictures.** The dots of the bold lines represent the mean scores.

report a change in affective state after completing the task. In contrast to our prediction (but in line with other, earlier studies reporting unsuccessful positive affect induction, as discussed

**Table 2. Affective state scores of participants in Study I and Study II.** Mean scores (with standard deviations), difference scores and range are displayed.

| Condition | Time | Spoken Descriptions of Pictures | Passively Viewing Pictures |
|---|---|---|---|
| | | M (SD) | M (SD) |
| Positive | Pre-test | 5.20 (0.88) | 4.94 (1.17) |
| | Post-test | 5.44 (0.83) | 5.06 (1.08) |
| | Difference | 0.24 (0.62) | 0.12 (0.86) |
| | Range | -1.50, 2.67 | -3.17, 2.17 |
| Neutral | Pre-test | 5.28 (0.89) | 5.13 (0.92) |
| | Post-test | 5.13 (0.89) | 4.92 (0.97) |
| | Difference | -0.15 (0.54) | -0.21 (0.36) |
| | Range | -1.33, 0.67 | -1.17, 0.50 |
| Negative | Pre-test | 5.25 (0.91) | 4.88 (1.03) |
| | Post-test | 3.97 (1.22) | 3.58 (1.12) |
| | Difference | -1.28 (1.17) | -1.30 (1.04) |
| | Range | -4.00, 0.67 | -3.83, 0.83 |

below), participants viewing and describing positive pictures did not report (significantly) higher levels of positive affect after the task.

## 3. Study II: Passively Viewing Pictures

Study II studied the effect of passively viewing, but not describing out loud, pictures gradually increasing in affective content on (self-reported) affective state. We used the same sets of pictures as in Study I.

### 3.1. Method

**3.1.1. Design.** The design was identical to Study I.

**3.1.2. Participants.** Participants were recruited at the same Dutch university as in Study I. A total of 126 participants participated in the experiment for course credit; none of them participated in Study I. Two participants did not consent to have their data published in scientific journals; therefore, we excluded their data. Our final sample included $N$ = 124 participants (43 male; $M$ age = 23.50, $SD$ age = 4.00), again, each assigned to one of the conditions (positive condition, $n$ = 41, neutral condition, $n$ = 41; negative condition, $n$ = 42). Again, all procedures were in accordance with the ethical standards of the local research committee. Written informed consent was obtained from all individual participants included in the study.

**3.1.3. Materials.** *Stimuli*. The materials we used were identical to those used in Study I, but in contrast to Study I, participants could do the experiment in Dutch or English, because they did not verbally describe the pictures, the language they spoke became irrelevant. Participants received informed consent, instructions and debriefing, and fill out the questionnaires, in their language of choice.

*Viewing IAPS pictures*. The viewing time per picture was identical to Study I.

*Procedure*. The procedure was identical to that of Study I, except that participants only passively viewed the pictures, instead of viewing them and describing them out loud. For this reason, no audio recording took place.

### 3.2. Results

**3.2.1. Descriptive statistics.** As in Fig 1, Fig 2 displays the individual scores of affective state (y-axis) for the pre- and post-test (x-axis), sorted by condition (positive, neutral, and negative pictures). On the y-axis, lower scores indicate the degree of unpleasant affect; higher scores indicate the degrees pleasant affect.

Notice that Fig 2 looks very similar to Fig 1, showing the same pattern as described above: participants viewing negative pictures generally reported feeling unpleasant, and participants viewing positive or neutral pictures generally did not report a substantial change in affective state. Akin to the participants in Study I, participants in Study II generally started the experiment in fairly good spirits, scoring roughly 5 to 5.5 on the 7-point Likert scale.

**3.2.2. Change in affective state.** A repeated measures analysis was performed with time (pre-test and post-test) as within-subjects factor, condition (positive, neutral or negative pictures) as between-subjects factor and affective state as dependent variable. Mean scores, standard deviations, difference scores and range can be found in Table 2. As in Study I, a main effect was found for time, $F (1, 121)$ = 40.22, $p < .001$, $\eta_p^2$ = .25, and condition, $F (2, 121)$ = 8.99, $p < .001$, $\eta_p^2$ = .13. However, again, these two main effects were qualified by a (predicted) interaction effect for time and condition, $F (2, 121)$ = 35.16, $p < .001$, $\eta_p^2$ = .37. Identical to Study I, post-hoc tests revealed that participants in the negative condition reported lower levels of pleasant state after viewing the negative pictures. Again, participants in the positive or

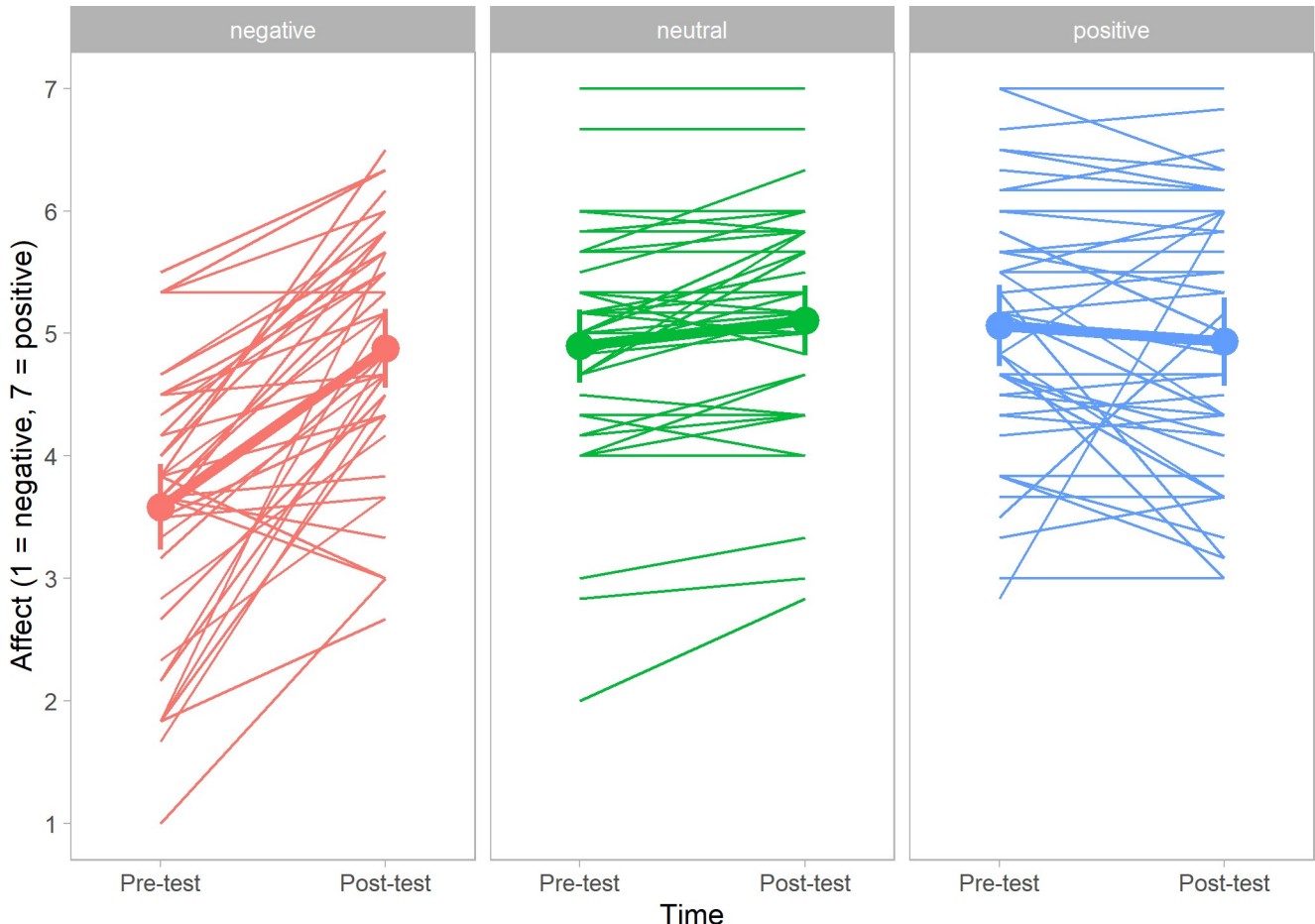

**Fig 2. Individual affect scores for participants Passively Viewing Pictures.** The dots of the bold lines represent the mean scores.

neutral condition did not report a significant difference in affective state after viewing the pictures.

### 3.3. Conclusion

Similar results to Study I were found: participants viewing negative pictures reported negative affect after viewing the pictures, and participants viewing positive or neutral pictures did not report a significant change in affective state.

## 4. Comparing Spoken Descriptions of Pictures and Passively Viewing Pictures

To determine if viewing and describing the positive and negative pictures out loud (Study I), compared to passively viewing them (Study II), evoked higher levels of (positive or negative, respectively) affect, the ratings from Study I and Study II were compared using an ANOVA.

### 4.1. Individual changes in affective state

To explore our dataset, we looked at the changes in affective state for all individual participants (both Study I and Study II). As can be inferred from both Fig 1 and Fig 2, there is a substantial

amount of variation in the effectiveness of the manipulation, with only the negative condition showing a consistent pattern for the majority of the participants.

Generally speaking, participants viewing positive pictures ($n$ = 82) reported feeling more pleasant after the task ($n$ = 47), albeit the change was modest ($\leq$ 1 on a 7 point scale) for the majority of participants ($n$ = 40). Regarding participants viewing neutral pictures ($n$ = 81), the majority ($n$ = 75) reported a small change in affective state, feeling more positive ($\leq$ 1) or negative ($\leq$ -1). Participants viewing negative pictures ($n$ = 82) showed the same pattern, but contrary to participants exposed to the neutral pictures, the variation between participants was much larger: 74 participants reported more unpleasant affect, of which 50 individuals reported a decrease of $\geq$ -1 on the affect scale.

There were no large (individual) differences between the individuals describing the pictures out loud, or only passively viewing them.

## 4.2. Results

To test the hypothesis that verbally describing affective pictures, compared to only viewing them, enhances the effect on affective state, a mixed ANOVA was performed with time (pre-test and post-test) as within-subjects factor, type of study (Study I: Spoken Descriptions of Pictures, or Study II: Passively Viewing Pictures) and condition (positive, neutral or negative pictures) as between-subjects factors, and affective state as dependent variable.

A main effect was found for type of study, $F$ (1, 239) = 6.39, $p$ = .012, $\eta_p^2$ = .03 (Study I: $M$ = 5.04, $SD$ = .91; Study II: $M$ = 4.75, $SD$ = 0.91), indicating that affective state was overall more positive for participants in Study I, compared to Study II. A main effect was also found for time, $F$ (1, 239) = 67.50, $p$ < .001, $\eta_p^2$ = .22 (pre-test: $M$ = 5.11, $SD$ = .97; post-test: $M$ = 4.68, $SD$ = 1.22), indicating that participants experienced more negative affect after engaging in the task (reflecting the effective manipulation in the negative condition). Finally, a main effect was found for condition, $F$ (2, 239) = 16.82, $p$ < .001, $\eta_p^2$ = .12 (positive: $M$ = 5.16, $SD$ = .94; neutral: $M$ = 5.11, $SD$ = .91; negative: $M$ = 4.42, $SD$ = .94), indicating that affect was lower overall in the negative condition, both for participants that described the pictures and for those that did not. However, no three-way interaction of time, type of study, and condition was found, $F$ (2, 239) = .09, $p$ = .915, indicating that describing or only viewing affective pictures did not influence affective state significantly for one or more of the conditions.

Given that there was no significant change in affective state for participants after viewing positive or neutral pictures, we wanted to rule out the possibility that these null effects obscured a possible difference for the negative condition. Selecting only the negative condition, a repeated measures ANOVA was performed with time (pre-test and post-test) as within-subjects factor, type of study (Study I or Study II) as between-subjects factor and affective state as dependent variable. As predicted, a main effect was found for time, $F$ (1, 80) = 111.95, $p$ < .001, $\eta_p^2$ = .583, with participants becoming more negative during the experiment. However, we found no effect for type of study, $F$ (1, 80) = 3.49, $p$ = .065, and, importantly, no interaction between time and study, $F$ (1, 80) = 0.001, $p$ = .98. The results of this secondary analysis again indicate that individuals experienced worse affective state after exposure to the pictures, regardless of whether they described the pictures out loud or not.

## 5. An exploratory content analysis of the picture descriptions

In order to investigate the language use of the participants, and get more insight in how individuals describe (affective) content, we explored the verbal picture descriptions of Study I, using the word counting software LIWC [2]. LIWC is a text analysis software program for counting words and calculating percentages of words, grouping them in various categories,

including cognitive- and affective processes. For our current analysis, we used the Dutch LIWC dictionary [30] to keep track of the words in the LIWC-categories 'affective processes' (to which we will refer to as "affective words", e.g., dirty, help), 'positive emotion' words (e.g., beautiful, hug), 'negative emotion' words (e.g., sad, cry), as well as the word count per picture description.

Verbalizations of descriptions were transcribed by five individuals outside the project. The utterances of $N = 122$ participants were transcribed, resulting in 40 x 122 = 4880 descriptions. Forty-three descriptions, less than 1% of the dataset, were missing: all descriptions from one participant (in the neutral condition), two descriptions from one participant, and one description from one participant. One participant was excluded because she did not consent to her data being used. Our final sample included 4797 picture descriptions by $n = 120$ speakers, with a mean word count of 18.89 ($SD = 6.73$) words per description.

## 5.1. Descriptives

Table 3 provides the mean percentages (with standard deviations) of total words used per picture description, in the corresponding LIWC categories, per condition. Fig 3A–3D depict the average scores per item in the respective LIWC category, represented by dots (the average score per item) and trend lines, including bands, representing Confidence Intervals. As can be seen in Fig 3A–3C, most individuals tend to use no (0) or few (1, 2) affective, positive, or negative emotion words to describe a picture. Participants gradually used more positive emotion words to describe positive pictures and negative pictures, but not neutral pictures (Fig 3A). The same pattern was found for negative emotion words, although the increase was less steep (Fig 3B). Affective word use gradually increased to describe positive and negative pictures, but not neutral pictures (Fig 3C). In all conditions, the data suggest that the number of words increases with subsequent pictures, a trend which is most clear for negative pictures (Fig 3D). However, we should be cautious interpreting this pattern since there is also substantial variation between participants.

## 5.2. Results

To statistically analyze word count and affective words used in the picture descriptions, four separate one-way ANOVAs were performed, with condition as independent factor, and word count, (percentage of) affective-, positive- and negative word use, as dependent variables. Data were aggregated on individual level, combining individual scores on each picture description to one mean score for each LIWC category. We tested for homogeneity of variances using Levene's tests. Results of ANOVAs and Levene's tests can be found in Table 4. Levene's test results indicated that equal variances were assumed for word count, but not for affective-, positive-, and negative word use. Differences between the conditions were assessed with Tukey's (equal variances assumed) and Games Howell (equal variances not assumed) post hoc comparisons.

**5.2.1. Affective words.**  Although our positive affect induction was not successful, participants describing positive pictures generally used more affective words in their descriptions, compared to participants describing neutral pictures, $p < .001$. Negative pictures were described with more affective words than neutral pictures, $p < .001$. No difference was observed for affective word use between positive and negative pictures, $p = .670$.

**5.2.2. Positive emotion words.**  Positive emotion words were used significantly more when describing positive pictures, compared to negative pictures, $p < .001$, and neutral pictures, $p < .001$. There was no significant difference between negative and neutral pictures, $p = .483$.

**Table 3. Mean scores, standard deviations and confidence intervals per condition for word count, and percentages of affective words, positive emotion words, and negative emotion words.**

| Condition | Descriptions $n$ | Affective words $M$ ($SD$) | CI | Pos. words $M$ ($SD$) | CI | Neg. words $M$ ($SD$) | CI | Word count $M$ ($SD$) | CI |
|---|---|---|---|---|---|---|---|---|---|
| Positive | 1638 | 1.54 (3.21) | 1.39–1.70 | 1.26 (2.85) | 1.13–1.40 | 0.13 (0.93) | 0.09–0.18 | 20.16 (6.52) | 19.84–20.47 |
| Neutral | 1560 | 0.75 (2.42) | 0.63–0.87 | 0.45 (1.98) | 0.35–0.55 | 0.27 (1.27) | 0.21–0.33 | 17.58 (6.90) | 17.24–17.92 |
| Negative | 1599 | 1.69 (3.43) | 1.52–1.86 | 0.53 (1.70) | 0.45–0.61 | 1.03 (2.76) | 0.90–1.17 | 18.88 (6.53) | 18.56–19.20 |
| Total | 4797 | 1.33 (3.08) | 1.25–1.42 | 0.75 (2.27) | 0.69–0.82 | 0.48 (1.87) | 0.43–0.53 | 18.89 (6.73) | 18.70–19.09 |

**5.2.3. Negative emotion words.** A similar pattern was observed for negative word use: participants describing negative pictures used significantly more negative emotion words, compared to positive pictures, $p < .001$, and neutral pictures, $p < .001$. Additionally, neutral pictures were described with more negative emotion words than positive pictures, $p = .012$.

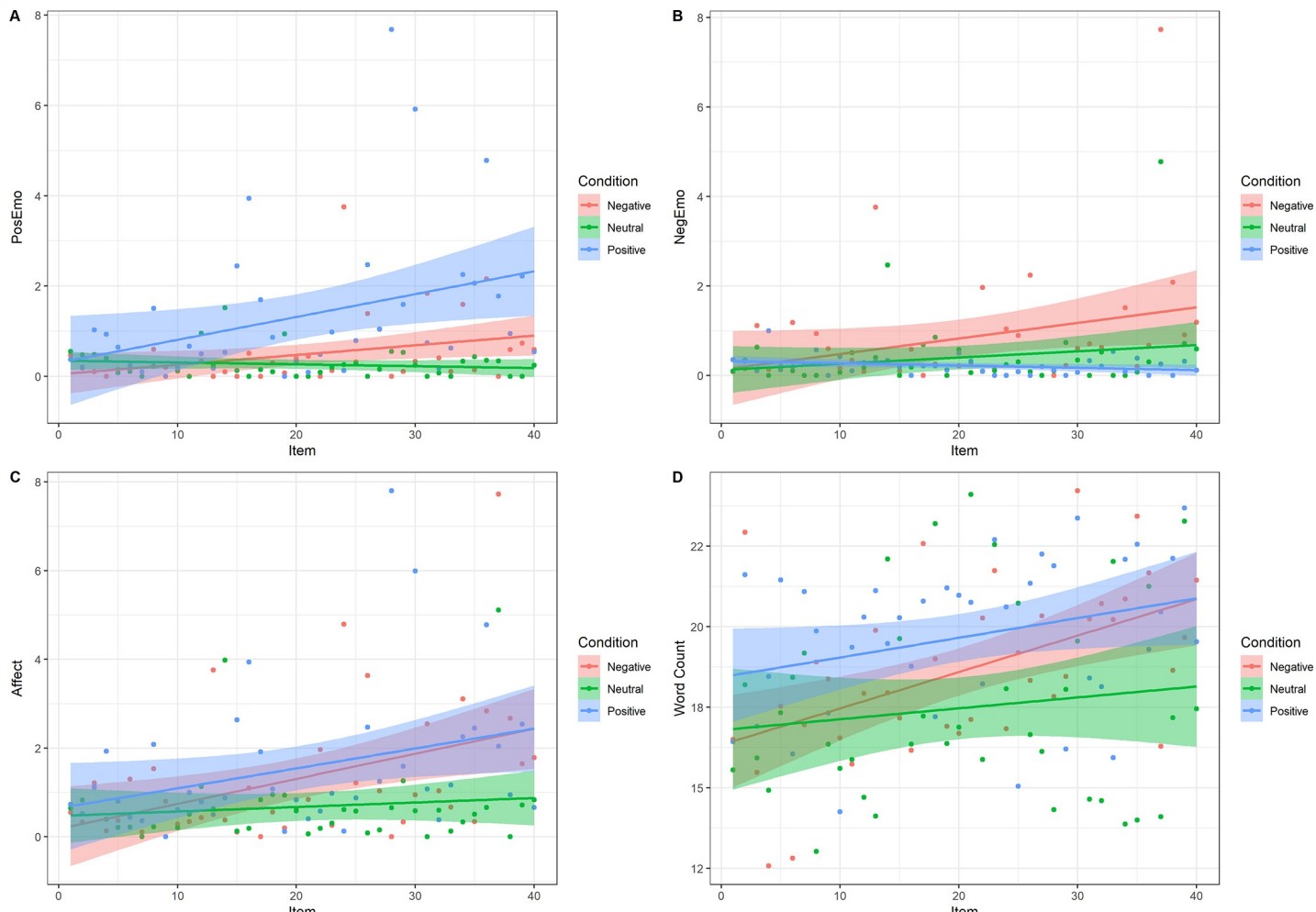

**Fig 3. Trend lines for average scores per item (represented by dots), per condition for positive emotion words, negative emotion words, affective words, and word count.** Bands represent confidence intervals.

Table 4. ANOVAs and Levene's tests for condition on affective-, positive-, and negative word use, and word count.

| | One-way ANOVA | Levene's test |
|---|---|---|
| Affective word use | $F$ (2, 117) = 20.78, $p$ < .001 | $F$ (2, 117) = 9.19, $p$ < .001 |
| Positive word use | $F$ (2, 117) = 25.57, $p$ < .001 | $F$ (2, 117) = 10.31, $p$ < .001 |
| Negative word use | $F$ (2, 117) = 134.87, $p$ < .001 | $F$ (2,117) = 6.77, $p$ = .002 |
| Word count | $F$ (2, 117) = 3.30, $p$ = .040 | $F$ (2, 117) = 0.22, $p$ = .801 |

**5.2.4. Word count.** Individuals used more words describing positive pictures compared to neutral pictures, $p$ = .031, but not compared to negative pictures, $p$ = .409. No significant difference was found between neutral and negative pictures, $p$ = .405

## 5.3. Conclusion

The results of this exploratory content analysis are in line with what would be intuitively expected. Individuals viewing affective pictures used more affective words in their descriptions, compared to when they are describing neutral pictures. Positive pictures, compared to negative and neutral pictures, were described with more positive emotion words, and conversely, negative pictures were described with more negative emotion words than neutral pictures, and neutral pictures were described with more negative emotion words than positive pictures. Speakers used more words to describe positive pictures than neutral pictures.

## 6. General discussion

In this study, we aimed to study the effect of free verbal expression on affect induction, by investigating the effectiveness of affect induction methods, inspired by Velten [1], where pre-defined self-referential statements are replaced with IAPS pictures, gradually increasing in affective content (positive, negative) or remaining neutral. In Study I, 'Spoken Descriptions of Pictures', individuals verbalized the content of the pictures out loud. In Study II, 'Passively Viewing Pictures', participants passively viewed the pictures, and did not describe them out loud. Our first hypothesis was partly confirmed: as predicted, for both studies, negative affect induction was effective, and the neutral condition did not evoke a change in affective state. However, in both studies, positive affect induction did not result in a significant enhancement of positive affective state when compared to the neutral condition. Our second hypothesis was not confirmed: describing the pictures out loud did not enhance, nor did it temper, affective state.

Additionally, the linguistic content of the verbal descriptions of the IAPS pictures was explored with LIWC [2]. For positive and negative pictures, we observed a gradual increase in affective word use over time. Specifically, positive pictures were described with more positive emotion words, and negative pictures were described with more negative emotion words. No effects were observed for the neutral pictures. A large variation between pictures was observed for the number of words speakers used to describe the pictures. In general, speakers used more words to describe positive pictures than neutral pictures, but not negative pictures.

### 6.1. Inducing positive and negative affect

Verbally describing and passively viewing affective pictures successfully induced negative affective states (in the negative condition), but not positive affective states (in the positive condition). Our findings did not support the hypothesis that verbally describing affective pictures would induce stronger affective states than passively viewing them. The finding that the

positive affect induction turned out to be less successful than the negative affect induction is found more often (e.g.,[21, 31, 32]), and other studies using IAPS pictures (e.g., [21]) and the Velten method (e.g. [31, 33–35]) have faced this problem as well. Given the fairly positive affective state of the participants before being exposed to the pictures, a possible explanation for this lack of an effect might be that the participants' positive affect was already at ceiling [21, 31].

## 6.2. Affect and language

Participants reported slightly more pleasant affect after describing the positive pictures, compared to passively viewing them. However, this difference was small and not significant. As described above, the literature shows mixed results regarding the effect of affect labelling on affective (and emotional) experience. Putting emotions into words can enhance the affective experience (e.g., [15]) or decrease it (e.g., [12]). However, we realize that our affect induction procedure is somewhat unique, and might not be directly comparable to affect labelling. First of all, in experiments studying affect labelling, participants are asked to describe their own affective state. In our Study I, individuals were asked to describe the (affective) content of affective pictures, not their own emotions. Second, our participants were asked to view and describe the pictures simultaneously, making it harder to distinguish between the effect of viewing the pictures and verbalizing the content. But given that the effect of verbalizing was small, we doubt whether it would have made a significant difference to first expose individuals to the pictures to our participants, and asking them to describe them only after viewing.

To date, only a limited amount of work has been done on the description of affective content where speakers could use their own words [36, 15]. Study I adds to this relatively new field that combines questions from affective science and psycholinguistics.

## 6.3. Verbal descriptions

Corresponding to the gradual increase in affective content of the positive and negative pictures, we observed a gradual increase in affective language use in the expected directions: over time, positive pictures were described with more affective and positive emotion words; negative pictures were described with more affective and negative emotion words. The descriptions of neutral pictures were described with few affective, positive- and negative emotion words. Interestingly, individuals viewing and describing positive pictures did not (self)report enhanced positive affect, but they *did* use substantially more positive emotion words in their descriptions, compared to the negative and neutral pictures.

The usage of emotional words could be attributed to and explained by the specific (affective) content of the pictures. Therefore, we also studied a phenomenon that could not be attributed to the emotional content of pictures—the general number of words uttered to describe the pictures. Our results indicated that individuals did not use more words to describe positive than negative pictures. This is in contrast to the literature: happy individuals tend to talk faster (e.g., [37], but see also [38]) and sad individuals tend to talk slower (e.g., [39]). Additionally, happy individuals have been found to utter more words spontaneously [1]. However, keeping in mind that positive affect induction was not successful, this finding is not unexpected. Another explanation might be that the content of the affective pictures was more complex, compared to the neutral pictures. Indeed, many neutral pictures included depictions of objects, patterns and portraits, whereas the affective pictures often composed scenes of multiple components, e.g., individuals in various situations (e.g., plane crash, cycling), diverse backgrounds (e.g., nature, city, living room).

However, individuals used more words to describe positive pictures than neutral pictures. Assuming that affective pictures, both positive and negative, are more arousing than neutral pictures, this might explain why individuals used more words to describe affective than neutral pictures, because highly aroused speakers compared to lowly aroused speakers tend to have an increased speech rate [40] and thus might use more words. We found that the IAPS arousal scores were indeed positively correlated to word use, both for positive (r = .07) and negative (r = .16) pictures. However, we also found correlations between the IAPS valence scores and word use, which were systematically larger than those between arousal and word use, for both the positive (r = .12) as well as negative pictures (r = -.21). Concluding, both affective and arousing content was correlated to the number of words used to describe the pictures.

## 6.4. Strengths and limitations

Our study has a few limitations that need to be acknowledged. First, while we deliberately chose to use an incremental procedure, the incremental nature of the affect induction procedure might pose various issues. Given that order effects are at the base of our study, participants might be influenced to a lesser extent by the pictures, because they were exposed to pictures gradually increasing in affective content, instead of viewing a random selection of affective pictures (e.g., [23, 41, 42]) that might be, on average, more positive or negative. The temporal place of a stimulus in an array of pictures can influence how the stimulus is processed in the viewer, e.g. habituation effects (e.g., [43]) might reduce the effectivity of the stimuli, while recency bias (e.g., [44]) might enhance the effect of the last (few) affective pictures. However, showing affective IAPS pictures in a fixed (or non-incremental) order is not uncommon, and has been shown to effectively induce desired emotions and affective states (e.g., [15, 45, 46]). Therefore, while we acknowledge this limitation, we do not think the incremental nature of the stimuli is responsible for the absence of an effect of, for example, the positive condition. Nevertheless, there is certainly a possibility that the last few pictures were the most effective at inducing affect, and the previous pictures' affective impact was limited. For future research, it might be interesting to compare the effectivity of exposure to highly positive or negative rated IAPS pictures, compared to exposure to pictures gradually increasing or decreasing in valence.

Based on the available evidence in the literature, predicting the precise effectivity of the incremental procedure was difficult. Hence, we were ambivalent in our predictions: the gradual increase in valence could result in a weaker effect, a stronger effect, of perhaps even no effect at all. For example, Van der Zwaag et al. [47] compared the effectivity of gradual versus abrupt change in happy music to sad music. They found that both emotion induction procedures were equally effective, lowering both valence and energy (i.e. feeling more tired, according to self-report of the participants).

Gradually increasing affective content of stimuli might have several advantages. First, as Velten argued, the gradual emotion induction was favorable, 'to overcome the subjects' presumable reluctance to experience unpleasant mood' [4] (p. 68). Indeed, recent research shows that noncompliance with an affect induction procedure is more common viewing negative videos than positive videos [48]. Given that we started the series of negative pictures with neutral stimuli, this might prevent the initial reluctance of participants to engage in the negative affect induction procedure. Additionally, for some populations, the startle effect might be specifically unethical, because they could cause serious psychological or physiological harm, for example, to individuals with certain mental disorders (e.g., PTSD, panic disorder) or cardio-vascular diseases.

Second, verbalizing the content of the pictures adds additional challenges–for example, participants likely vary in their degree of verbal skills and consequently differ in how difficult they

considered the task. We did try to take this into account in the selection of our participants by excluding participants with a speech disorder or a limitation in the ability to speak fluently (e.g., stuttering). However, to check whether having Dutch as a first language had an effect on the effectivity of the affect induction of verbally describing the pictures, we repeated our analysis of Study I, excluding the participants who did not have Dutch as their first language ($n = 5$), but did not find substantial differences. Additionally, given that we tested a relatively homogenous group of participants (young Dutch students), we expect that individual differences in verbal fluency, attention, and other cognitive and communicative abilities are small and randomly distributed throughout our sample.

Third, an additional benefit of our study is the collection of human, realistic verbal descriptions for the content of the subset of IAPS pictures we used. While these descriptions are not yet validated, it is a valuable first step to the possible creation of a verbal IAPS, which might be useful in certain specific populations, e.g., visually impaired individuals.

Lastly, we conjectured that asking individuals to use their own words describing the pictures (instead of uttering pre-defined affective sentences), would reduce the awareness of the goal of the procedure (affect induction) and therefore reduce the chance of participants reporting to feel the change in affect they think they 'should' experience (e.g., social desirability or task demands), even when they do not actually experience a shift in affective states (e.g., [49]).

## 6.5. Future research

Affective processes often take place in a social setting, but in laboratory settings, they are generally induced in individual participants [50]. Our affect induction method might be a useful, naturalistic method to induce affective states in more than one individual at the same time. For example, participants could take turns in a dialogue setting describing out loud the affective IAPS pictures to each other. This might create more naturalistic opportunities in affective research to study affect induction in dyads.

Contrary to the Velten method, our method describing pictures was not self-referential in nature. Recent literature suggests that self-referencing might play a critical role in affective word processing. Soares et al. [51] found that in a masked priming paradigm, individuals categorize positive adjectives faster when they are primed by self-related primes, compared to other-primes. In light of these findings, the Velten method might be more effective inducing positive affect than our affect induction method. Upon inspecting the verbalizations, indeed, only 26% percent of the descriptions are self-referential (e.g., 'I see. . .'), and less than 1% is other-referential ('Here you see. . .'; see S3 Table). This might be one of the reasons that our pleasant affect induction was not effective. For future research, it might be interesting to compare a condition where participants are instructed to provide a self-referential description ('I see a happy couple') to a condition where participants are instructed to provide a non-self-referential description of the pictures ('This is a picture of a happy couple').

Finally, our affect induction method is not inherently limited to valence, but also could be applied to specific emotion categories that are present in picture datasets (cf., [22], for IAPS). By replacing the current pictures with pictures that induce a specific emotion (e.g., disgust, tenderness and anger), we think our method might be able to successfully induce specific emotions and their accompanying verbal descriptions.

## 6.6. Implications

This study contributes to the sparse literature on verbalizing affective content, implying that an engaging task as verbalizing negative content, using free expression, can be an effective method to induce negative affect in a possibly more ecologically sound manner (e.g., viewing

affective videos). The results indicated that verbalizing or passively viewing affective content are equally effective methods to induce negative affective state.

We contributed to the scientific literature on the relationship between affect and language, aiming to gain understanding of the critical, but unclear relationship between language production and (un)pleasant affect.

## Supporting information

**S1 Table. Positive and negative IAPS pictures that were initially selected but excluded based on our exclusion criteria.**
(DOCX)

**S2 Table. Final selection of positive (increasing in valence), negative (decreasing in valence), and neutral IAPS pictures.**
(DOCX)

**S3 Table. Self-referential and other-referencing in the pictures descriptions, sorted by condition, counted by LIWC.**
(DOCX)

**S1 Fig. Valence (y-axis) of IAPS pictures by bin (x-axis).**
(TIF)

**S2 Fig. Arousal (y-axis) of IAPS pictures by bin (x-axis).**
(TIF)

## Author Contributions

**Conceptualization:** Charlotte Out, Martijn Goudbeek, Emiel Krahmer.

**Data curation:** Charlotte Out.

**Formal analysis:** Charlotte Out, Martijn Goudbeek.

**Funding acquisition:** Martijn Goudbeek, Emiel Krahmer.

**Investigation:** Charlotte Out.

**Methodology:** Charlotte Out.

**Project administration:** Charlotte Out.

**Supervision:** Martijn Goudbeek, Emiel Krahmer.

**Validation:** Charlotte Out, Martijn Goudbeek, Emiel Krahmer.

**Visualization:** Charlotte Out, Martijn Goudbeek.

**Writing – original draft:** Charlotte Out.

**Writing – review & editing:** Charlotte Out, Martijn Goudbeek, Emiel Krahmer.

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
