## [Decision Letter · Decision Letter 0]

29 Aug 2019

PONE-D-19-17931

Gradual positive and negative emotion induction using images: the effect of verbalizing emotional content

PLOS ONE

Dear Mrs. Out,

Thank you for submitting your manuscript to PLOS ONE. After careful consideration, we feel that it has merit but does not fully meet PLOS ONE’s publication criteria as it currently stands. Therefore, we invite you to submit a revised version of the manuscript that addresses the points raised during the review process.

Based on mainly Reviewer 1's comments I encourage a revision of you manuscript focussing on those aspects of framing.

We would appreciate receiving your revised manuscript by Oct 13 2019 11:59PM. To enhance the reproducibility of your results, we recommend that if applicable you deposit your laboratory protocols in protocols.io, where a protocol can be assigned its own identifier (DOI) such that it can be cited independently in the future. For instructions see: http://journals.plos.org/plosone/s/submission-guidelines#loc-laboratory-protocols

We look forward to receiving your revised manuscript.

Kind regards,

Hedwig Eisenbarth

Academic Editor

PLOS ONE

Journal Requirements:

2. Please provide additional details regarding participant consent. If your study included minors, state whether you obtained consent from parents or guardians. If the need for consent was waived by the ethics committee, please include this information.

3. We noted in your submission details that a portion of your manuscript may have been presented or published elsewhere: "Some results in the current manuscript have been published in an abstract for the CERE 2018 conference. This abstract focuses on the main effect of emotion induction on affective state of the participants. However, in the current manuscript, we also:

1) studied the effectiveness of the emotion induction on individual level, instead of only group level

2) compared our data to another study using the Velten method (Wilting et al., 2006)

3) Included tables and figures that are not in the CERE abstract

4) Discussed the literature on emotion and language production in more depth.

Additionally, the abstract is barely 1.5 pages, compared to our current manuscript of about 30 pages."

Reviewers' comments:

Reviewer's Responses to Questions

**Comments to the Author**

1. Is the manuscript technically sound, and do the data support the conclusions?

Reviewer #1: Yes

Reviewer #2: Yes

2. Has the statistical analysis been performed appropriately and rigorously? 

Reviewer #1: Yes

Reviewer #2: Yes

3. Have the authors made all data underlying the findings in their manuscript fully available?

Reviewer #1: Yes

Reviewer #2: Yes

4. Is the manuscript presented in an intelligible fashion and written in standard English?

Reviewer #1: Yes

Reviewer #2: Yes

5. Review Comments to the Author

Reviewer #1: Review of Gradual positive and negative emotion induction using images: the effect of verbalizing emotional content (PONE-D-19-17931)

In this manuscript, the authors conducted two emotion induction experiments inspired by the Velten method. Using stimuli from IAPS, the authors placed participants in one of three emotion induction conditions: negative, positive, or neutral, where images either increased in negative valence, increased in positive valence, or remained neutral. In the first experiment, participants saw each image and described what they saw for 10 seconds in one of the three valence conditions. In the second experiment, participants passively viewed the images for 10 seconds each. The authors reported successful emotion induction, most successful for the negative condition. They found the induction results comparable to the Velten method.

The methods, analyses, and conclusions were sound. My main issue with the manuscript is the novelty of the question. To me this does not demonstrate a new method of emotion induction. Rather, it introduces additional complications that may not be necessary. One critical test of the method’s novelty would be to ask why an experimenter who wants to induce negative emotions would use this method rather than showing a random set of highly negative IAPS pictures?

The gradual increasing of affective images might complicate emotion induction, such as imposing order effects and taking longer (400 seconds in this case). Alternatively, other studies have randomized blocks of negative images distributed throughout a task to remove order effects and to sustain negative affect induction. By using one long series of images, it is also possible that participants are just more responsive to the last portion of the most negative images, or that the most negative images at the end are made less salient because of participants are slowly ramped up to them.

Describing while viewing emotional images adds another layer of complication, such as attention and verbal skills. This may be more relevant as a comparison to other induction methods, or for explicitly testing the influence of verbalizing when exposed to emotional stimuli, which the authors partially reviewed but was not the intended question.

To clarify my issue, for example, if a gradual increasing method showed a more sustained or a different kind of negative affect induction, it would demonstrate novelty. Alternatively, if the authors could demonstrate specific behavioral consequences of this emotion induction method, or a particular reason for which a gradual increase of valence or the description of emotional stimuli is useful, or a particular set of questions or dependent measures that are differentially targeted by description of pictures rather than passive viewing, it would demonstrate novelty.

Other notes:

-what was the dependent variable of the ANOVAs? If it was valence ratings, using a magnitude change may reveal a greater and the interaction effect smaller. If it's not a magnitude change, if both negative and positive inductions were perfectly and equally effective, you would hypothesize a null time effect and predict only the interaction.

-the other component of the Velten method is the self-referential nature of the stimuli. There is now a literature on the impact of affective words when they influence the self vs. another. If the main thrust of the question is a direct comparison with the Velten method, this would ideally be addressed.

-despite comparable ratings, the positive valence IAPS pictures are sometimes considered to be not as significant as the negative valence pictures, possibly related to the greater ease in inducing negative emotions (which might account for the reduced positive induction). Also within the negative pictures, I’ve noticed the disgust images are more effective inducers than the fear pictures.

-Figure 3 should be bar plots. Line suggests an ordinal or continuous relationship between the three experiments

-the comparison of the emotion induction results with another study that used the Velten method could be placed in the Results section.

Reviewer #2: This is a sound and well presented study. It builds on earlier work on the elicitation of changes in emotional states via linguistic stimuli, and presents the results of two experiments involving visual stimuli. However, a limited amount of space is devoted to a discussion of the findings. I suggest that the authors include more explanation of the findings and more comments on their implications. It should also be clarified why the use of images, as opposed to linguistic stimuli, is described as more 'natural'.

6. PLOS authors have the option to publish the peer review history of their article (what does this mean?). If published, this will include your full peer review and any attached files.

Reviewer #1: No

Reviewer #2: No

---

## [Author Response · Author response to Decision Letter 0]

3 Mar 2020

Reviewer #1

Review of Gradual positive and negative emotion induction using images: the effect of verbalizing emotional content (PONE-D-19-17931)

In this manuscript, the authors conducted two emotion induction experiments inspired by the Velten method. Using stimuli from IAPS, the authors placed participants in one of three emotion induction conditions: negative, positive, or neutral, where images either increased in negative valence, increased in positive valence, or remained neutral. In the first experiment, participants saw each image and described what they saw for 10 seconds in one of the three valence conditions. In the second experiment, participants passively viewed the images for 10 seconds each. The authors reported successful emotion induction, most successful for the negative condition. They found the induction results comparable to the Velten method.

The methods, analyses, and conclusions were sound. My main issue with the manuscript is the novelty of the question. To me this does not demonstrate a new method of emotion induction. Rather, it introduces additional complications that may not be necessary. One critical test of the method’s novelty would be to ask why an experimenter who wants to induce negative emotions would use this method rather than showing a random set of highly negative IAPS pictures?

We thank the reviewer for their insightful and valid comments. We understand the main issue highlighted by the reviewer.

We realize now that we probably have not been clear enough in describing our aims and would like to clarify these below. The primary motivation was to study the effect of free verbal expression on emotion induction. Given that the relationship between language production and emotion induction is not well understood, we set out to investigate whether verbally describing emotional images would result in an enhanced emotional experience, compared to passively viewing the images. For this, we decided to pick a relatively natural and unconstrained approach, by presenting individuals with emotional IAPS images, asking them to describe the content freely.

To bring out this focus more clearly in the revised manuscript, we have adapted the title and we have now included a section of the paper exploring the content of participant’s verbal descriptions of the IAPS images. These descriptions were transcribed, and analyzed using Linguistic Inquiry and Word Count (LIWC), focusing on the production of emotional words as a function of the three conditions (positive, neutral, and negative), and over time (from the first image to the last image).We describe these in the Results section of a new subsection, ‘Exploratory analyses: Content analysis of image descriptions’.

A secondary motivation of our study is indeed exploring alternative emotion induction methods. As highlighted by, for example, Schmidt et al. (2018), most emotion induction methods are passive and not very ecologically valid. In our study, participants actively engaged in the induction method, especially in the setting where they were asked to verbally describe the images in such a way that other people would be able to identify the described image. Instead of reading out loud fixed statements (as in the Velten method), participants were asked to think for themselves and use their own words. We think this contributes to the need for more ecologically sound emotion induction methods. One attractive feature of the Velten method is its incremental nature. Somewhat surprisingly, this unique property of the procedure has not been studied or discussed in the literature, leaving us wondering whether the incremental manner of inducing mood is also effective when using (IAPS) images with affective content.

In the new manuscript, we now describe our focus more clearly in the Introduction.

The gradual increasing of affective images might complicate emotion induction, such as imposing order effects and taking longer (400 seconds in this case). Alternatively, other studies have randomized blocks of negative images distributed throughout a task to remove order effects and to sustain negative affect induction. By using one long series of images, it is also possible that participants are just more responsive to the last portion of the most negative images, or that the most negative images at the end are made less salient because participants are slowly ramped up to them.

Thank you for this comment. Indeed, for emotion induction purposes, it is common to present IAPS images in a random order (e.g., Capecelatro et al., 2013; Dhaka & Kashyap, 2017). 

We conjecture that gradually increasing emotional valence of stimuli might have several advantages. First, the first stimulus is unlikely to shock or startle participants. When participants are startled by images, they might feel excessively tense and want to quit engaging in the study, which is both detrimental to the quality and quantity of the research, as well as ethically questionable. Velten argued in favor of gradual emotion induction, ‘to overcome the subjects’ presumable reluctance to experience unpleasant mood’ (Velten, 1967, p.68). Additionally, for some populations, the startle effect might be specifically unethical, because they could cause serious psychological or physiological harm, for example, individuals with certain mental disorders (e.g., PTSD, panic disorder), cardiovascular diseases, and the elderly. 

Incidentally, we were unsure what the reviewer referred to with ‘taking 400 seconds longer’. We reckoned that they referred to the Spoken Descriptions of Images study, where we state that each image was viewed for 10 seconds, resulting in 400 seconds of viewing images (40 images x 10 seconds). We did not include information on the viewing time of the images in the Passively Viewing Images study, which might have been confusing. The viewing time per image was identical in both studies. We have now added this information to Passively Viewing Images study (Materials).

Describing while viewing emotional images adds another layer of complication, such as attention and verbal skills. This may be more relevant as a comparison to other induction methods, or for explicitly testing the influence of verbalizing when exposed to emotional stimuli, which the authors partially reviewed but was not the intended question.

First, we would like to clarify that we did intend to explicitly test the influence of verbalizing when exposed to emotional stimuli (as we described in response to the first comment from this reviewer). One of our initial hypotheses was that describing emotional images would enhance the effect on emotional state compared to passively viewing the images. We compared the two methods and concluded that there was no substantial difference in effect on mood. 

We agree that attention and verbal skills could play a role in our ‘Describing and viewing images’ study. We did consider this, for example, by defining relevant exclusion criterion, such as having a speech disorder or limitation in the ability to speak fluently (for example, stuttering). Moreover, given that we tested a relatively homogenous group, we expect that other variabilities (cognitive and communicative abilities) will be randomly distributed throughout the sample, and therefore not have a major impact on our results. 

We have reframed the revised manuscript, focusing more on the effect of verbalizing on affective state. We have added a new section, ‘Exploratory analyses: Content analysis of image descriptions’, exploring the content of the verbal descriptions of the images using LIWC, and also briefly discuss these results in the General Discussion.

To clarify my issue, for example, if a gradual increasing method showed a more sustained or a different kind of negative affect induction, it would demonstrate novelty. Alternatively, if the authors could demonstrate specific behavioral consequences of this emotion induction method, or a particular reason for which a gradual increase of valence or the description of emotional stimuli is useful, or a particular set of questions or dependent measures that are differentially targeted by description of pictures rather than passive viewing, it would demonstrate novelty.

We understand the reviewer’s point. As described above, in the revised manuscript we describe more explicitly that our primary interest is to study the effect of spoken language production on affective state. To demonstrate the novelty of our study more clearly, we have now transcribed and analyzed the verbal descriptions of the images using LIWC, which offers an additional dependent measure. Additionally, we have strengthened our discussion of the possible uses of our approach as an emotion induction method.

Other notes:

What was the dependent variable of the ANOVAs? If it was valence ratings, using a magnitude change may reveal a greater and the interaction effect smaller. If it's not a magnitude change, if both negative and positive inductions were perfectly and equally effective, you would hypothesize a null time effect and predict only the interaction.

In our analysis, we used a composite score (which we called "Affect") based on the 6 scales introduced in Mackie and Worth. This score ranges from 1 (very negative) to 7 (very positive) (see Materials for details). We have added this information now to ‘Change in emotional state' (both 2.2.2. and 3.2.2.). We agree with the reviewers interpretation of the possible effects; if both the negative and positive mood inductions were equally effective, we would get a null effect of mood induction (because the effects of the positive and negative condition cancel each other out) and only get an interaction between time and mood (because over time, the scores for the positive and the negative condition do start to diverge). However, as discussed in both Results sections (2.2 and 3.2), in our studies, only the negative mood induction was found to be successful.

-the other component of the Velten method is the self-referential nature of the stimuli. There is now a literature on the impact of affective words when they influence the self vs. another. If the main thrust of the question is a direct comparison with the Velten method, this would ideally be addressed.

The reviewer has a good point; indeed, our study and the Velten method differentiate with respect to self-referencing. Even though our main focus is on language production (as explained above), we do feel it is interesting to address this topic. We now briefly discuss recent work on this (such as Soares et al., 2019). When inspecting the picture descriptions, we find that only a small percentage (26%) is self-referential (e.g., ’I see…’), as opposed to many utterances in Velten’s original method. This might partially explain why our positive emotion induction was not effective. We now briefly touch upon this in the General Discussion (‘Future research’). For future research, it might be interesting to compare a self-referential condition (‘I see a happy couple’) to a non-self-referential condition of verbal image descriptions (‘This is a picture of a happy couple’). We have included this suggestion in the Future Research section.

-despite comparable ratings, the positive valence IAPS pictures are sometimes considered to be not as significant as the negative valence pictures, possibly related to the greater ease in inducing negative emotions (which might account for the reduced positive induction). Also within the negative pictures, I’ve noticed the disgust images are more effective inducers than the fear pictures.

We thank the reviewer for this clear observation. Indeed, this is common problem inducing positive emotions with various emotion induction procedures, and we refer to this problem in the General Discussion. We now briefly touch upon this issue regarding the effectivity of the positive valence images of the IAPS in specific. 

-Figure 3 should be bar plots. Line suggests an ordinal or continuous relationship between the three experiments

The reviewer is, of course, correct. We have replaced Figure 3 with a bar plot. 

- the comparison of the emotion induction results with another study that used the Velten method could be placed in the Results section.

We have moved the comparison with the Wilting study to ‘5. Comparing our results to the original Velten method’.

Reviewer #2

This is a sound and well presented study. It builds on earlier work on the elicitation of changes in emotional states via linguistic stimuli, and presents the results of two experiments involving visual stimuli. 

However, a limited amount of space is devoted to a discussion of the findings. I suggest that the authors include more explanation of the findings and more comments on their implications. 

It should also be clarified why the use of images, as opposed to linguistic stimuli, is described as more 'natural'.

We thank the reviewer for their compliments, and agree with their suggestions. We have expanded our General Discussion section, now explicitly discussing the limitations and strengths of our study. We discuss the possibility of order effects, the additional challenges of the verbalization of the stimuli, and how our ‘Describing Images’ study contributes to the need for more ecological and engaging mood induction procedures. We have rewritten the future research section, and added an implications section. We have also added new material, now dedicating a section of the paper to the analysis of the content of the verbal descriptions of the IAPS images (‘Exploratory analyses: content analysis of image descriptions’). 

We also explain in somewhat more detail what we mean when we claim that our emotion induction method is more natural than the Velten method. This is because participants in our study were able to use their own words to describe affective stimuli, as opposed to the Velten method in which they read out loud fixed affective statements.

---

## [Decision Letter · Decision Letter 1]

30 Mar 2020

PONE-D-19-17931R1

The effect of language on emotion: verbalizing images gradually increasing in emotional content

PLOS ONE

Dear Mrs. Out,

Sorry for the delay with getting back to you. Unfortunately we did not have the chance to obtain reviewer of the two reviewers who had previously reviewed your manuscript. However, we were lucky to find two experts in the field who were able  to review your revised manuscript, also in light of earlier comments and the original version of your manuscript.

Thank you for submitting your manuscript to PLOS ONE. After careful consideration, we feel that it has merit but does not fully meet PLOS ONE’s publication criteria as it currently stands. Therefore, we invite you to submit a revised version of the manuscript that addresses the points raised during the review process.

The comments of reviewer 3 might be new but seem highly valuable in terms of consistent wording and conceptualisation of emotion, valence and arousal. Those clarifications along side the suggested edits will definitely strengthen your manuscript, therefore I suggest you to follow those points.

We would appreciate receiving your revised manuscript by May 14 2020 11:59PM. To enhance the reproducibility of your results, we recommend that if applicable you deposit your laboratory protocols in protocols.io, where a protocol can be assigned its own identifier (DOI) such that it can be cited independently in the future. For instructions see: http://journals.plos.org/plosone/s/submission-guidelines#loc-laboratory-protocols

We look forward to receiving your revised manuscript.

Kind regards,

Hedwig Eisenbarth

Academic Editor

PLOS ONE

Reviewers' comments:

Reviewer's Responses to Questions

**Comments to the Author**

1. If the authors have adequately addressed your comments raised in a previous round of review and you feel that this manuscript is now acceptable for publication, you may indicate that here to bypass the “Comments to the Author” section, enter your conflict of interest statement in the “Confidential to Editor” section, and submit your "Accept" recommendation.

Reviewer #3: (No Response)

Reviewer #4: All comments have been addressed

2. Is the manuscript technically sound, and do the data support the conclusions?

Reviewer #3: Partly

Reviewer #4: Partly

3. Has the statistical analysis been performed appropriately and rigorously? 

Reviewer #3: No

Reviewer #4: Yes

4. Have the authors made all data underlying the findings in their manuscript fully available?

Reviewer #3: Yes

Reviewer #4: Yes

5. Is the manuscript presented in an intelligible fashion and written in standard English?

Reviewer #3: Yes

Reviewer #4: Yes

6. Review Comments to the Author

Reviewer #3: This paper introduces new data that can be brought to bear on existing debates about the influence of language on (the intensity of) affective experience. The authors have made some valuable changes and additions in their revisions - in particular, the examination of the language used to describe the images. Yet there are still some points where clarifications and perhaps new analyses are needed. Please note that, although I did not serve as a reviewer during the first round, I read both versions of the manuscript before preparing my comments on the revision.

1. I would be cautious with the use of the word ‘emotion(al)’ throughout the manuscript. What is being induced and measured is a change in affect, valence, or un/pleasantness. The ‘affective content’ of the stimuli is therefore a more appropriate description. Likewise, I would refer to an ‘affect induction’ rather than an ‘emotion induction’, to ‘affective language’ rather than ‘emotional language’, etc.

2. At several points in the manuscript, the authors appear to conflate valence, arousal, and intensity (e.g. page 9 line 221). In relation to the selection of stimuli, I would also like to see the authors comment on how the “strong increase in arousal for negative images” (page 9, line 226) does or does not impact their results and conclusions. Could it be that this arousal difference is driving the negative affect induction? Relatedly, the authors suggest that positive and negative images may be more arousing than neutral images, which may explain differences in the number of words used in image descriptions (page 24, line 569). This conjecture could be directly tested.

3. Can the authors clarify why they analyzed pre- and post-induction ratings separately, rather than creating change scores? It seems that change scores would preserve individual differences while streamlining the analyses and narrative. If the authors believe change scores are not appropriate, then I would highly recommend transforming the affect ratings to be centered on 0, such that negative ratings indicate negative affect, and positive ratings indicate positive affect. This will greatly assist in the interpretation of results and figures. For example, lines 276-8 (“lower scores indicate higher levels of negative emotion; higher scores indicate higher levels of positive emotion”) would be much easier to follow if scores were centered on 0.

4. There appears to be a slight mis-interpretation of the main effects reported on page 16 of the results. On lines 385-6, the authors state that “describing images enhanced emotional state in the participants”. To the contrary, a main effect of study type merely indicates that affect was more pleasant overall in Study 1 – this effect alone does not indicate an effect of the induction. Similarly, on lines 390-1, the authors state that “the emotion manipulation was effective for individuals exposed to negative images”. A main effect of condition simply means that affect was lower overall in the negative condition; causality cannot be inferred without examining change in affect.

5. Much of the authors’ argument throughout the manuscript hinges on the effectiveness of having participants verbally describe the evocative images. Yet this argument isn’t always supported by the analyses and results. For example, in comparing the present studies with the work of Wilting et al. (pages 17-8), the authors only analyze the post-induction affect ratings for all 3 studies. Without taking pre-induction ratings into account, however, this comparison isn’t meaningful (i.e. without controlling for where participants started, we don’t know how much they were influenced by the task). Similarly, the authors state that “verbally describing emotional images is an effective method to induce, especially negative, emotional states” (page 22, lines 525-6), while the results indicate no effect of study (page 17, lines 401-2). Also, the claim that “participants generally reported stronger emotions after describing emotional images compared to only viewing them” (page 23, lines 533-4) conflates valence with intensity – higher ratings indicate more pleasant affect, which actually works against the effectiveness of a negative affect induction. These aspects should be revised to clarify the findings and contribution of the present work.

6. Miscellaneous:

a. I prefer the original title, as the revised title seems to place undue emphasis on the effect of language on emotion (see above), and is also easy to misread as “verbalizing images gradually increases emotional content”

b. The original version of the manuscript included a brief description of the Velten method. This seems to have been removed in the revision, and I think it should be added back to help readers such as myself who were not otherwise familiar with the method.

c. Page 5, line 123: “especially for individuals who are free to use their own words” – I would rephrase as “especially when individuals are allowed to use their own words”

d. “Self-referral” (e.g. page 3, line 68) should be “self-referring” or “self-referential”

e. Page 9, line 212: what does it mean that the neutral images were pseudo-randomly selected?

f. Page 10, line 236: “electing” should be “eliciting”

g. Page 20 line 469: “Data were aggregated” (data is a plural noun)

h. Page 22, line 530: I would replace “happiness” with “positive affect”

Reviewer #4: The Authors did addressed all the points raised by reviewer 1. However, some of the authors' responses raise more questions. In particular, regarding the claims of incremental mood induction being "ecologically superior", the authors also pointed out that incremental induction of emotion has not been tested before on affective content images. Thus, only further investigations will support their statement of ecological validity. I suggest deleting lines 615-624 of the draft, and reference to "ecological sound" as a suggestion and not as a fact (lines 658-660 of the draft).

7. PLOS authors have the option to publish the peer review history of their article (what does this mean?). If published, this will include your full peer review and any attached files.

Reviewer #3: No

Reviewer #4: No

---

## [Author Response · Author response to Decision Letter 1]

28 Apr 2020

Dear Dr. Eisenbarth,

Please find enclosed a revision of our manuscript “Gradual positive and negative affect induction: the effect of verbalizing affective content” (PONE-D-19-17931R1). We would like to thank you and the two reviewers of our revised manuscript for the time and consideration. We were happy to read that reviewer #4 feels all initial comments have been adequately addressed, and agree that the last comments and suggestions from reviewer #3 are valuable in their attention to consistent wording and the clear conceptualisation of emotion, valence and arousal. Below we describe how all separate comments by reviewer #3 and #4 were dealt with.

We hope to have addressed all comments and suggestions appropriately, and that the current revision will be judged favourably as a contribution to PLOS One.

Kind regards,

Charlotte Out, Emiel Krahmer and Martijn Goudbeek

Reviewer #3

This paper introduces new data that can be brought to bear on existing debates about the influence of language on (the intensity of) affective experience. The authors have made some valuable changes and additions in their revisions - in particular, the examination of the language used to describe the images. Yet there are still some points where clarifications and perhaps new analyses are needed. Please note that, although I did not serve as a reviewer during the first round, I read both versions of the manuscript before preparing my comments on the revision.

We thank the reviewer for their comments and compliments.

1. I would be cautious with the use of the word ‘emotion(al)’ throughout the manuscript. What is being induced and measured is a change in affect, valence, or un/pleasantness. The ‘affective content’ of the stimuli is therefore a more appropriate description. Likewise, I would refer to an ‘affect induction’ rather than an ‘emotion induction’, to ‘affective language’ rather than ‘emotional language’, etc.

The reviewer is correct and we have changed this accordingly.

2. At several points in the manuscript, the authors appear to conflate valence, arousal, and intensity (e.g. page 9 line 221). In relation to the selection of stimuli, I would also like to see the authors comment on how the “strong increase in arousal for negative images” (page 9, line 226) does or does not impact their results and conclusions. Could it be that this arousal difference is driving the negative affect induction? Relatedly, the authors suggest that positive and negative images may be more arousing than neutral images, which may explain differences in the number of words used in image descriptions (page 24, line 569). This conjecture could be directly tested.

Regarding the conflation of valence, arousal and intensity, we have rewritten the manuscript, carefully considering the appropriate use of these terms. For example, we have changed line 221 to ‘affective content’ instead of ‘intensity’.

Indeed, the strong increase in arousal for negative images might impact our results and conclusions. To control for this, we followed the suggestion of the reviewer, now also looking at the correlations between the IAPS valence and arousal scores for the positive and negative images and the average number of words used by participants to describe these images. The results indicate that both valence and arousal were correlated to the number of words used to describe the images in the affective conditions, however, correlations between word use and valence were substantially higher. We describe these results in the Discussion.

3. Can the authors clarify why they analyzed pre- and post-induction ratings separately, rather than creating change scores? It seems that change scores would preserve individual differences while streamlining the analyses and narrative. If the authors believe change scores are not appropriate, then I would highly recommend transforming the affect ratings to be centered on 0, such that negative ratings indicate negative affect, and positive ratings indicate positive affect. This will greatly assist in the interpretation of results and figures. For example, lines 276-8 (“lower scores indicate higher levels of negative emotion; higher scores indicate higher levels of positive emotion”) would be much easier to follow if scores were centered on 0.

To some extent, this might to be a matter of subjective preference. Given that the effect of time (pre- and posttest) is important for answering our hypothesis (does affective state change after viewing/describing positive, neutral or negative images?), we prefer the repeated measures analyses keeping both the pre- and posttest scores, because we feel the results are more intuitive and easier to interpret than change scores. Change scores also make it harder to see how participants actually reported to feel in the different conditions. 

However, we understand the concern of the reviewer and agree that the results sections could be improved to make them more easy to interpret. Therefore, we have now added change scores to Table 2, and clarified our explanations of the results by streamlining our language following the suggestions of the reviewer.

4. There appears to be a slight mis-interpretation of the main effects reported on page 16 of the results. On lines 385-6, the authors state that “describing images enhanced emotional state in the participants”. To the contrary, a main effect of study type merely indicates that affect was more pleasant overall in Study 1 – this effect alone does not indicate an effect of the induction. Similarly, on lines 390-1, the authors state that “the emotion manipulation was effective for individuals exposed to negative images”. A main effect of condition simply means that affect was lower overall in the negative condition; causality cannot be inferred without examining change in affect.

The reviewer is correct; thank you for spotting this. Regarding the first comment, we have changed this to “(…) indicating that affective state was overall more positive for participants in Study I, compared to Study II”. Regarding the second comment, we have changed this to “(…) indicating that affect was lower overall in the negative condition, both for participants 

that described the images and for those that did not.’

5. Much of the authors’ argument throughout the manuscript hinges on the effectiveness of having participants verbally describe the evocative images. Yet this argument isn’t always supported by the analyses and results. For example, in comparing the present studies with the work of Wilting et al. (pages 17-8), the authors only analyze the post-induction affect ratings for all 3 studies. Without taking pre-induction ratings into account, however, this comparison isn’t meaningful (i.e. without controlling for where participants started, we don’t know how much they were influenced by the task). 

Unfortunately, the pre-test scores of the Wilting et al. study were never collected. We understand the reviewer’s concern and have therefore decided to removed this comparison.

Similarly, the authors state that “verbally describing emotional images is an effective method to induce, especially negative, emotional states” (page 22, lines 525-6), while the results indicate no effect of study (page 17, lines 401-2). 

We have rephrased this sentence to “Verbally describing and passively viewing affective pictures successfully induced negative affective states (in the negative condition), but not positive affective states (in the positive condition). Our findings did not support the hypothesis that verbally describing affective pictures would induce stronger affective states than passively viewing them.’

Also, the claim that “participants generally reported stronger emotions after describing emotional images compared to only viewing them” (page 23, lines 533-4) conflates valence with intensity – higher ratings indicate more pleasant affect, which actually works against the effectiveness of a negative affect induction. These aspects should be revised to clarify the findings and contribution of the present work.

We have changed this sentence to ‘Participants reported slightly more positive affect after describing the positive images, compared to passively viewing them. However, this difference was small and not significant.’

6. Miscellaneous:

a. I prefer the original title, as the revised title seems to place undue emphasis on the effect of language on emotion (see above), and is also easy to misread as “verbalizing images gradually increases emotional content”

b. The original version of the manuscript included a brief description of the Velten method. This seems to have been removed in the revision, and I think it should be added back to help readers such as myself who were not otherwise familiar with the method.

c. Page 5, line 123: “especially for individuals who are free to use their own words” – I would rephrase as “especially when individuals are allowed to use their own words”

d. “Self-referral” (e.g. page 3, line 68) should be “self-referring” or “self-referential”

e. Page 9, line 212: what does it mean that the neutral images were pseudo-randomly selected?

f. Page 10, line 236: “electing” should be “eliciting”

g. Page 20 line 469: “Data were aggregated” (data is a plural noun)

h. Page 22, line 530: I would replace “happiness” with “positive affect”

We have changed the title to “Gradual positive and negative affect induction by describing versus viewing affective pictures’’. Moreover, we have added a brief description of the original Velten method to the Introduction. The small errors have been corrected.

Regarding e), we understand the confusion and changed ‘pseudo-random’ to ‘random’, because neutral images were selected in a similar fashion as the affective images, i.e. at random. 

Reviewer #4: 

The Authors did addressed all the points raised by reviewer 1. However, some of the authors' responses raise more questions. In particular, regarding the claims of incremental mood induction being "ecologically superior", the authors also pointed out that incremental induction of emotion has not been tested before on affective content images. Thus, only further investigations will support their statement of ecological validity. I suggest deleting lines 615-624 of the draft, and reference to "ecological sound" as a suggestion and not as a fact (lines 658-660 of the draft).

We thank the reviewer for this observation and understand the need for clarification.

We have substantially rephrased lines 615-624, now suggesting that the engaging nature of the method is a potential strength, and we have removed mentioning of ecological superiority. In the Implications section of the Discussion, we now suggest that our method is possibibly more ecologically sound, instead of stating this as a fact.

---

## [Decision Letter · Decision Letter 2]

11 May 2020

Gradual positive and negative affect induction: the effect of verbalizing affective content

PONE-D-19-17931R2

Dear Dr. Out,

We are pleased to inform you that your manuscript has been judged scientifically suitable for publication and will be formally accepted for publication once it complies with all outstanding technical requirements.

With kind regards,

Hedwig Eisenbarth

Academic Editor

PLOS ONE

Additional Editor Comments (optional):

Reviewers' comments:

Reviewer's Responses to Questions

**Comments to the Author**

1. If the authors have adequately addressed your comments raised in a previous round of review and you feel that this manuscript is now acceptable for publication, you may indicate that here to bypass the “Comments to the Author” section, enter your conflict of interest statement in the “Confidential to Editor” section, and submit your "Accept" recommendation.

Reviewer #3: All comments have been addressed

Reviewer #4: All comments have been addressed

2. Is the manuscript technically sound, and do the data support the conclusions?

Reviewer #3: Yes

Reviewer #4: Yes

3. Has the statistical analysis been performed appropriately and rigorously? 

Reviewer #3: Yes

Reviewer #4: Yes

4. Have the authors made all data underlying the findings in their manuscript fully available?

Reviewer #3: Yes

Reviewer #4: Yes

5. Is the manuscript presented in an intelligible fashion and written in standard English?

Reviewer #3: Yes

Reviewer #4: Yes

6. Review Comments to the Author

Reviewer #3: The authors have addressed my comments on the manuscript. I thank them for their responsive and attentive revisions. I especially appreciate the streamlined analyses and new title.

Reviewer #4: The authors have met the reviewers' recommendation . I consider the paper now ready for publication.

7. PLOS authors have the option to publish the peer review history of their article (what does this mean?). If published, this will include your full peer review and any attached files.

Reviewer #3: No

Reviewer #4: No

---

## [Editor Report · Acceptance letter]

18 May 2020

PONE-D-19-17931R2 

Gradual positive and negative affect induction: the effect of verbalizing affective content 

Dear Dr. Out:

I am pleased to inform you that your manuscript has been deemed suitable for publication in PLOS ONE. Congratulations! Your manuscript is now with our production department. 

With kind regards,

on behalf of

Dr. Hedwig Eisenbarth 

Academic Editor

PLOS ONE